# Urban Noise and Psychological Distress: A Systematic Review

**DOI:** 10.3390/ijerph17186621

**Published:** 2020-09-11

**Authors:** Nicola Mucci, Veronica Traversini, Chiara Lorini, Simone De Sio, Raymond P. Galea, Guglielmo Bonaccorsi, Giulio Arcangeli

**Affiliations:** 1Department of Experimental and Clinical Medicine, University of Florence, 50139 Florence, Italy; nicola.mucci@unifi.it (N.M.); giulio.arcangeli@unifi.it (G.A.); 2Occupational Medicine School, University of Florence, 50139 Florence, Italy; 3Department of Health Sciences, Hygiene and Preventive Medicine, University of Florence, 50139 Florence, Italy; chiara.lorini@unifi.it (C.L.); guglielmo.bonaccorsi@unifi.it (G.B.); 4Department of Anatomical, Histological, Forensic and Locomotor Apparatus Sciences, Sapienza University of Rome, 5 Piazzale Aldo Moro, I-00185 Rome, Italy; simone.desio@uniroma1.it; 5Faculty of Medicine & Surgery, University of Malta, MSD 2080 Msida, Malta; raymond.galea@um.edu.mt; 6Head of the Malta Postgraduate Medical Training Programme, Mater Dei Hospital Msida, MSD 2090 L-Imsida, Malta

**Keywords:** urban noise, environmental, annoyance, sleep disorders, health disorders, residents, exposure, dose–response

## Abstract

Chronic exposure to urban noise is harmful for auditory perception, cardiovascular, gastrointestinal and nervous systems, while also causing psychological annoyance. Around 25% of the EU population experience a deterioration in the quality of life due to annoyance and about 5–15% suffer from sleep disorders, with many disability-adjusted life years (DALYs) lost annually. This systematic review highlights the main sources of urban noise, the relevant principal clinical disorders and the most effected countries. This review included articles published on the major databases (PubMed, Cochrane Library, Scopus), using a combination of some keywords. The online search yielded 265 references; after selection, the authors have analyzed 54 articles (5 reviews and 49 original articles). From the analysis, among the sources of exposure, we found the majority of items dealing with airports and wind turbines, followed by roads and trains; the main disorders that were investigated in different populations dealt with annoyance and sleep disorders, sometimes associated with cardiovascular symptoms. Regarding countries, studies were published from all over the world with a slight prevalence from Western Europe. Considering these fundamental health consequences, research needs to be extended in such a way as to include new sources of noise and new technologies, to ensure a health promotion system and to reduce the risk of residents being exposed.

## 1. Introduction

Noise pollution is defined as “noise in the living environment or in the external environment such as to cause discomfort or disturbance to rest and human activities, danger to health, deterioration of ecosystems, material goods, monuments, the external environment or such as to interfere with use of the rooms themselves” [1]. This type of pollution can mainly result from vehicle traffic, railways, airports, constructions, industries, recreational activities, etc. [2]. Worldwide many people are exposed to this risk factor and they can suffer the relative consequences. In Western Europe, at least one million healthy life years are lost per year [3]. Actually, as many as 125 million European citizens are exposed to road traffic, which is above the average annual levels of 55 dB, however, these figures could actually be significantly higher. Such an exposure causes a perception of annoyance for 20 million inhabitants. In 8 million inhabitants, sleep disorders appear and causes more than 40,000 hospitalizations. In addition, around 8000 children in Europe are believed to have difficulty in reading and with concentrating in areas where air traffic noise is close to school buildings [4]. Prolonged exposure to noise can be harmful to the auditory perception, with the onset of perceptual hearing loss, and to other human systems, in particular the cardiovascular, gastro-enteric, and nervous systems; it can also cause psychological annoyance, defined by ISO/TS 15666:2003(E) through the expression “one person’s individual adverse reaction to noise” [5]. Road traffic noise can lead to the development of cardiovascular and metabolic disease [6,7] and possibly oncological disorders [8,9]. Additionally, this exposure may increase the risk of weight gain [10], obesity [11,12] and Type II diabetes mellitus [13]. Data on the possible development of oncological pathologies are still controversial; some studies on urban noise demonstrated a positive association between these exposures and breast cancer [14]; on the other hand, other studies found no association [8]. A case-control study carried out on women found no association between cancer and traffic or railway noise, but a positive association with aircraft-noise exposure [9]. Prolonged negative feelings towards noise may increase the risk of more severe psychological problems [15]. It has been shown, through very well documented subjective data, that annoyance and sleep disturbances are the most widespread reported disorders associated with environmental noise [16]. Tiredness, headaches and other psychological conditions are also associated with noise in adult populations [17,18,19]. Psychological distress has been recognized as a substantial public health problem and as a leading cause of morbidity and disability [20]. It accounts for most of the community burden of poor mental health [21]. It has been estimated that around 25% of the EU population experience a deterioration in quality of life due to annoyance, and about 5–15% suffer from sleep disorders [22]. In fact, according to WHO, most of lost disability-adjusted life years (DALYs) can be attributed to noise-induced sleep disturbance and annoyance [3]. Because of this, the EU has issued directives on the subject. The 2002/49/CE Directive has the primary objective of avoiding, preventing or reducing the harmful effects of exposure to environmental noise, by determining the exposure to noise (by means of acoustic mapping), public information on noise’ effects and the adoption of action plans [23]. In addition, Legislative Decree 194/2005 implements the previous directive on the determination and management of environmental noise; it defines the procedures of competences for the installation of strategic noise maps in urban areas with more than 100,000 inhabitants, guaranteeing public participation [24]. This systematic review aims to identify the sources of urban noise that cause the most discomfort to residents, the main psychological disorders associated with the condition and the countries which are most effected.

## 2. Materials and Methods

This systematic review follows the Prisma Statement [25].

### 2.1. Literature Research

The research included articles published in the last 10 years, from 2010 to 29 February 2020, on the major online databases (Pubmed, Cochrane Library and Scopus). The search strategy used a combination of controlled vocabulary and free text terms based on the following keywords: noise, annoyance, exposure, dose–response. All research fields were considered. Additionally, we carried out a manual search on reference lists of the selected articles and reviews, so as to carry out a wider analysis. Two independent reviewers read the titles and abstracts of the reports that were identified by the search strategy. They selected the relevant reports according to the inclusion and exclusion criteria. Doubts or disagreements were solved by discussing the issue with a third researcher. Subsequently, they individually screened the corresponding full text, so as to be able to decide on final eligibility. Finally, the authors eliminated any duplicate studies and articles where full texts were not available. Data was mainly obtained from the published results but also from any other supplementary sources when these were available. In particular, the authors have selected date of publication, country of examined residents, number of included residents, questionnaire administered, the involved source of noise, exposure decibel and the type of disturbance reported. In addition, the authors highlighted the number of studies included for all reviews and the length of the experiment in the case of trial or cohort studies.

### 2.2. Eligibility and Inclusion Criteria

The studies included in this review focus on urban noise and the residents that are exposed to this risk. Articles on exposure to major sources of urban noise such as airports, railways, roads and wind turbines were included. We have only included studies concerning psychological disorders, in particular annoyance and sleep disorders. All types of study designs were included. No restrictions were applied either by language or country.

### 2.3. Exclusion Criteria

Reports related only to occupational exposure, publications on programmatic interventions and studies not related to psychological disorders were excluded. Additionally, reports of less academic significance, editorial articles, individual contributions and purely descriptive studies published in scientific conferences without any quantitative and qualitative inferences were excluded.

### 2.4. Quality Assessment

Three different reviewers assessed the methodological quality of the selected studies with specific rating tools, to reduce risk of introducing any bias (Table 1). We used the International Narrative Systematic Assessment (INSA) method to judge the quality of the narrative reviews [26], Assessment of multiple systematic reviews (AMSTAR) to evaluate systematic reviews [27] and the Newcastle Ottawa Scale (NOS) to evaluate cross-sectional, cohort studies and case control studies [28]. The Jadad Scale was applied for randomized clinical trials [29]. In addition, to reduce risk of bias, we have used RobVis (BARR, Bristol, UK), a specific tool for systematic reviews [30] (see Appendix A).

## 3. Results

The online research yielded 265 studies: PubMed (60), Scopus (186) e Cochrane Library (19). Of these, 128 studies were excluded because they were deemed not to be related to problems associated with urban noise. Of the remaining, 40 articles were also excluded because they were duplicates. Duplicate publications were carefully eliminated in order not to introduce bias by comparing the authors’ names, the issues addressed, workers’ destinations and the results obtained. Another 43 publications were eliminated because full text was not available. 54 studies were finally included in this systematic review (Figure 1). Of these, 2 were systematic reviews, 3 were narrative reviews and 49 were original articles. Among these original articles, 41 were cross-sectional studies, 3 cohort studies, 3 case-control studies and 2 trials (Table 2). Germany is the country in which most studies have been published (10 articles; 18.5%). Most of the articles were published in 2017 (10 studies; 18.5%), followed by 2016 and 2019 (9 and 8 articles, respectively; 16.6% and 14.8%). The selected articles investigate mainly the psychological distress’ symptoms experienced by residents, such as annoyance (28 studies; 51.8%), sleep disorders (11 articles; 20.3%) or both (11 articles; 20.3%). When taking into account the studies that examine a single source of noise, it was found that airport noise was the prevalent exposure that was reported on (15 articles; 27.7%), followed by road traffic, wind turbines and railways (10, 8 and 4 studies; 18.5%, 14.8% and 7.4%, respectively).

### 3.1. Narrative and Systematic Reviews

Regarding the methodological quality of the selected reviews, the AMSTAR score shows an average of 7, thus indicating a discrete quality of the studies (Table 3). The most appropriate methodological systematic review was conducted in Germany by WHO (AMSTAR = 8). Regarding the narrative reviews scores, the INSA score shows an average of 5.6, a median and a modal value of 6, indicating an intermediate quality.

Each review addresses different topics, both regarding the source of noise and the pathology that was investigated. Annoyance and sleep disturbance were reported to be more frequent near wind turbines than other sound sources, especially in rural areas. Annoyance has been reported with sound exposure above 40 dBA. Regarding sleep disturbances, these occur at higher sound levels, above 45 dB and this problem is significantly related to annoyance [31]. Hume highlights that this alteration appears from 30–40 dB at exposure near airports at night. New technologies will play an ever greater and more important role such as in the case of “open rotor engine”, which may improve over the coming 10–20 years, becoming significantly more fuel efficient, producing less carbon dioxide per air mile, but generating more noise [32].

Hays reviews the scientific literature on oil and gas development activities. This economic sector generates low frequency noise (for example, by compressor stations) but only limited data exists regarding the consequences, such as cardiovascular risks or adverse birth outcomes. Most of these activities are not permanently located in technological areas, so there may be fewer studies on the possible long-term effects [33]. Potential cardiovascular risk was also investigated by Lercher, in the Alpine Region. He focused on two studies, the Noise Village Study and the Transit Study, in both of which no relevant relationship between traffic noise and systolic blood pressure was demonstrated. The authors have highlighted a possible linear relationship with systolic pressure but, only in men, over 60 years and exposure to sound levels between 50 and 60 dBA Lden (OR = 1.38, CI = 1.03–1.86) [34]. Guski has described the association between exposure to various environmental noises and annoyance. He found that the relationship between noise levels and annoyance is stronger for noise generated by aircraft and railway than for road traffic and wind turbines. The rate of annoyed people is elevated in both “high-rate change” airports, such as Frankfurt and Berlin-Brandenburg, and “low-175 rate change” airports, such as Cologne/Bonn and Stuttgart [35].

### 3.2. Original Articles

The scores assigned to the original articles have an average value of 6.2, a median of 6 and a modal of 6 (Table 4). These numbers point to an intermediate quality of the studies with research from Switzerland, Netherland, France, Sweden and Austria obtaining the highest values (NEW CASTLE = 8).

In order to carry out the results and considered the quantity of the selected articles, we proceed with a synthesis of the results based on the urban noise sources and main disorders found by the authors.

#### 3.2.1. Sources of Noise

There were four main sources of exposures to noise investigated by the authors; 13 articles (13/47; 27.6%) investigate only noise from airport sources, seven from damage caused by wind turbines (7/47; 14.8%), nine from road or motorway traffic (9/47; 19.1%) and 4 from rail traffic (4/47; 8.5%). In 13/47 articles (27.6%), multiple sources were involved, seven studies dealt mainly with noise generated by airport/train/road (7/13; 53.8%), three dealt with road/rail generated noise (3/13; 23%), two (2/13; 15.3%) with airport/road and one (1/13; 7.6%) with wind turbine/airport. We found that the type of airport could influence the symptoms reported by the population effected. Morinaga found that living near military airports has a worse consequence than living near civilian infrastructures. In fact, comparing his data with a survey on civil airports, the author shows that more decibels are needed to obtain the same values of highly annoyed people [36]. The percentage of insomnia and sleep disorders vary with the increase of night flight operations [37]. In addition, Mueller found that the average of “awakeness” decreased from two in 2011 to 0.8 in 2012 due to the fact that there were less night flights [38]. Schreckenberg, in 2016, showed how levels of annoyance and sleep disorders decreased after some interventions in the airport. This did not affect disturbance upon awakening in the early morning [39]. There was a correlation between “value at which half of the people in a community describe themselves as highly annoyed by noise exposure” (CTL) and number of aircraft movements. In fact, near high rate of change (HRC) airports, the authors found more annoyed people. Gjestland found that 20% of the sample is highly annoyed when exposed to 55 db in the vicinity of HRC airports; on the other hand, close to the “low rate of change” airports, only 5% were annoyed when exposed to the same decibels [40]. Similarly, Silva has showed that the air traffic at Guarulhos airport increased about 45% on the last 5 years before the survey, as well as the percentage of annoyed citizens [41]. The location of the dwelling also has an effect on the annoyance. This symptom at particular sites with sea wave sound was significantly lower than that at sites without, probably because of noise masking by sea wave sounds [42]. In the Schreckenberg study, researchers found that residents suffered more sleep disturbance due to railway noise even when windows were closed (*p* < 0.001), and this was independent of the type of fixtures (soundproof windows, single-/double-glazed windows) [43]. Concerning road traffic, the association between L Night (overall night noise level) and these disturbances was dependent on the orientation of the bedroom to the nearest street. It was shown that when a bedroom pointed away from the nearest street, sleep disorders were less [44]. As far as the relationship between the distance to the noise source and the prevalence of annoyance, some researchers highlight that the rate of annoyed people rapidly decreased when moving away from the railway tracks. Ragettli found that highly annoyed people comprised 22% within 50 m, 10% within 51–100 m, and below 10% when the distance was in excess of 100 m from major roads [45]. Similarly, indoor noise annoyance was systematically reduced with increasing distance from wind turbines. In the data provided by Hongisto, the rate of annoyed people was around 10% when within 1200 m of such noisy sources, becoming negligible when about 2 km away from the source [46]. Annoyance and sleep are also influenced by other factors. In the Schreckenberg study, the individual noise sensitivity was correlated with aircraft noise annoyance (r = 0.36) but not with the sound level. Annoyance was higher in the group of middle-aged adults (40–60 years) when compared to younger or older people (*p* < 0.001). It was also higher in the middle to higher socio-economic status group (*p* < 0.001) and in house owners (*p* < 0.001). The fear of diminished house value was correlated with this disorder (r = 0.54, *p* < 0.001) [47]. Pedersen found that when just one stressor was operational respondents attributed noise and odor as the main annoying factor in 51% and 27% of cases respectively. When more than one stressor was present it was found that 32% were sensitive to noise, 43% to odor, and 32% to vibrations [48]. Sensitivity was shown to be a significant modifying factor (*p* = 0 in railway and roads) while gender was significant for railway noise (*p* = 0.014), as it pertains to subjective sleep disturbance [49]. Brown found that medium and high noise sensitivity categories were 1.5- and 2.4-times more likely to be highly annoyed. This was particularly so amongst residents who were not satisfied with their neighborhoods. These showed a 3.5-times more likelihood to be highly annoyed [50]. When Ogren compared vibration exposure to noise exposure from railway, traffic the noise levels and vibration velocities appeared to have the same probability of causing annoyance. For equivalent noise level and vibration, the probability of annoyance is approximately 20% for 59 dB or 0.48 mm/s, and about 40% for 63 dB or 0.98 mm/s. He found that annoyance from noise may be influenced by the presence of vibration (*p* = 0.022), but annoyance from vibration is perhaps not influenced so much by the noise level (*p* = 0.72) [51]. In 2019, Brink hypothesized that highly intermittent noise has an increased potential to disturb human activities. He confirmed that highly intermittent rail and aircraft noise interfere with annoyance level, but there was an opposite effect about road traffic noise: in latter, exposure with low intermittent noise (such as motorways) was associated with “highly annoyed” responses [52].

#### 3.2.2. Main Disorders

Of the 47 original articles included, 28 exclusively investigate annoyance (28/47; 59.5%). In the other cases, nine publications focused their findings on sleep disorders (9/47; 19.1%) while as many as nine articles investigated both disorders, both annoyance and sleep disorders. Finally, in four cases (4/47; 8.5%), in addition to the psychological domain, cardiovascular disorders due to urban noise were also reported. Of the 28 studies that exclusively investigate annoyance, 6 correlate this disorder with both airport and road noise (6/28; 21.4% respectively). From amongst the nine exclusive studies on sleep disorders, three correlate to wind turbines, two to aircraft, one to road, one to rail, one to road/rail and one to airport/rail/road. Ancona estimated that levels higher than 55 dB cause more than 4000 cases of hypertension and more than 9000 of annoyance. In the areas where night levels reach 50 dB, there were over 5000 sleep disorder events [53]. In Poland, health burden due to noise was caused by the annoyance (49%), sleep disturbance (38%) and ischemic heart diseases (13%) The author estimated that annoyance was causing 12,000 mean DALYs [54]. The most important contributor to the Sweden disease burden was sleep disturbances, accounting for 22,218 DALYs (54%), followed by annoyance with 12,090 DALYs (30%) and cardiovascular diseases with 6725 DALYs (16%) [55]. In Germany, the highest burden was attributable to road traffic noise, with 75.896 DALYs [56]. For Kim, the prevalence of sleep disturbance was high in the order of noise level (*p* < 0.001). The mean scores of the PSQI subscale were high, increasing with the level of noise, except in the case of sleep latency and use of sleeping drugs [57]. In Poulsen’s study, nocturnal noise exposure over 42 dB was associated with a hazard ratio (HR) of 1.14 (CI: 0.98–1.33) for sleep medication and HR of 1.17 (CI: 1.01–1.35) for antidepressants. The association was strongest amongst people over 65 years, with HR of 1.68 (1.27–2.21) for sleep medication and of 1.23 (0.90–1.69) for antidepressants [58]. In addition, Lercher investigated the relationship between railway noise and sleep medication intake; he showed that there was a doubled probability of medication intake at any level of railway sound exposure, with a statistically significant levelling off at around 60 dB [59]. Problems related to insomnia are often found mostly in noise-sensitive individuals and those interested in environmental issues [60]. Sleep disturbances are mostly found at levels above 45 dB. This correlation was significant in quiet areas (r = 0.208, *p* < 0.05) and also in quiet and noisy areas (r = 0.160, *p* < 0.01) [61]. The most annoyed had a lower mean domain for all HRQOL domains than those not annoyed, in particular physical (*p* < 0.001), psychological (*p* < 0.001), social (*p* < 0.001) and environmental domains (*p* < 0.001) [62]. Some authors have observed an association between aircraft noise annoyance and psychological distress, with a ORs of 4.00 (CI 1.67–9.55) for extremely annoyed people [63]. Only one article included in this research assessed the effects of construction site noise on residents. Liu found that this problem affects mental activities and sleeping more than watching TV or listening to music, more so in the morning (*p* < 0.05) [64]. Exposure–response relationships for waking, falling asleep, conversation, telephone listening, TV/radio listening, reading/thinking, and rest disturbances was found also in the study conducted by Shimoyama [65]. Some authors found that more than half of the respondents felt particularly annoyed in the late evening hours (between 20–23 h). Additionally, at a level of 60 dBA the model predicts 14% of highly annoyed respondents at daytime increasing to 36% during the evening, and 39% during the night-time period. Railway noise caused a variety of reactions in exposed residents, such as closing of windows, or feelings of anger or irritableness or conversation/radio louder [66]. Fryd has found differences between motorways and urban ways. In the case of motorways when the noise level was Lden 58 dB, 22% were highly annoyed while 48% were annoyed, as opposed to 8% who were highly annoyed and 28% who were annoyed in the case of urban roads. Comparing highly annoyed respondents in both types of roads, it is clear that 20% of those exposed to motorways when compared to urban roads were highly annoyed when exposed to a 10 dB decrease in noise level (55–60 dB vs. 65–70 dB). There is thus an important difference in outdoor annoyance such as in motorway case, where the respondents were more annoyed with less dB) [67].

#### 3.2.3. Countries

In 7 studies, the research involved exposed areas in Germany (7/47; 14.8%), 6 cases come from Japan, 5 from Austria, 4 from Sweden, 3 from the USA, 2 each from Italy, Switzerland, China, Netherlands, Denmark, Korea, Norway and 1 case from France, Thailand, Arab, Vietnam, Canada, Poland, New Zealand and Brazil. Among the seven studies from Germany, four investigated the airport environments (4/7; 57.1%); this was also the case for Japan with three studies (3/6; 50%). Of the 5 Austrian studies, 4 focus on trains and roads, particularly in the Alpine region, on the border with the Brenner.

### 3.3. Trials

We have found only two experimental studies (2/47; 4.2%) (Table 5). Comparing three different laboratory experiments on how sleep is effected by noise, Elmenhorst found that different noise sources produce different consequences. At the same decibel, the awakening probability was highest with exposure to railways, followed by exposure to road traffic and airport noise. However, the awakening probability from road traffic and railway noise is not significantly different (*p* = 0.988) [68]. In 2015, Schmidt tested the effects of nocturnal aircraft noise on cardiovascular function in 60 patients, between the ages of 30 and 75 years. The team simulated noises in the patients’ bedroom, producing 60 events during one night; they recorded polysomnography, endothelial function by flow-mediated dilation of the brachial artery and blood sampling on the next morning. The researchers found that sleep quality was markedly reduced by noise (from 5.8 ± 2.0 to 3.7 ± 2.2) (*p* 0.001), flow mediate dilatation significantly reduced (from 9.6 ± 4.3 to 7.9 ± 3.7%; *p* 0.001) and systolic blood pressure was increased (from 129.5 ± 16.5 to 133.6 ± 17.9 mmHg; *p* = 0.030). However, the adverse vascular effects of noise were independent from sleep quality and self-reported noise sensitivity [69].

## 4. Discussion

Noise has negative consequences for the health of exposed individuals. This is widely documented in the scientific literature [70,71,72]. Thus, increased blood pressure and cardiovascular disorders are associated with chronic exposure to noise, especially if originating from an airport [73,74,75,76,77]. In addition to the extra-hearing damage, there is a subjective alteration generally referred to as “noise disorder” or “annoyance” [78]. This arises when a sound source is perceived as annoying, irritating, unwanted, and associated with the presence of symptoms such as irritableness, fatigue, headaches, decreased performance, etc. Noise, similarly to other stressors, can activate the sympathetic nervous system [79], with consequent increase in heart rate and blood pressure, vasoconstriction, changes in blood viscosity, blood lipids and electrolyte alterations [80]. Prolonged exposure to noise can lead, in the most susceptible individuals, to permanent damage, ranging from hypertension to ischemic diseases, to myocardial infarct [81,82] and stroke [73]. Effects such as immune system dysfunction [83], psychological alterations such as irritability, aggressiveness, and decreased cognitive performance (e.g., difficulty understanding written language) have also been observed in individuals exposed to airport noise [84].

Our review has highlighted some specific risk factors present in this environmental sector, which are deserving of adequate consideration, in particular the prevention of repercussions on the health of residents. As can be expected, most studies agree that annoyance depends on the level of exposure; in fact, a higher exposure increases the rate of annoyed people. In the literature, the association between noise exposure and noise annoyance has been extensively investigated; aircraft noise has been found to be the most annoying among all transportation sources [85]. Recent research suggests that annoyance due to aircraft noise has increased over the previous years [86,87,88,89]. Noise emanating from vibrating movements and with a spectral content in low frequencies, (such as aerial noise), leads to noise reactions that are much more evident than other types of noise, such as tachycardia [90]. In this review it is evident that the disturbance most reported is annoyance, in relation to airports and road traffic. This disorder is linked to very variable factors such as the number of landings and take-offs, the type of aircraft used, the procedures and routes used at these stages and, of course, the characteristics of the territory at the take-off and landing routes besides the density of population and human activities. In fact, to protect the environmental quality, from an acoustic point of view, a rather complex regulatory system is in place, which includes Community Directives and Regulations, national and regional regulations of implementation, technical standards, involving, in the collegiate body constituted by the Airports Commissions, various subjects: technical-management (ENAC, ENAV, Airport Management Company), institutional (Ministry of the Environment, Region), local authorities (Communities and Provinces), carriers (airline representatives) [91].

Concerning vehicle traffic noise, which has a certain continuity and repetitiveness, it seems that the predominant effect is on sleep disturbance [82,92]. For this reason, WHO suggests that outdoor sound events with levels greater than 45 dBA should be avoided for a healthy night rest. In addition, background noise one meter from the exterior of the bedroom must not exceed 45 dB (A), in order to keep the windows open at night [93]. Other authors also found negative effects of noise on nocturnal rest. These have shown an increased risk of getting up tired and not rested in the morning [94], an increased motility and heart rate [95] and pseudo-neurological complaints (palpitation, heat flushes, dizziness, anxiety and depression) [96]. Noise induced disturbances vary according to the physical characteristics of the noise events [97]; in fact, dose–response relationships between night sound levels of aircraft noise and effects on sleep could be substantially improved by adding the number of noise events [98]. In addition, Saremi indicated that for the same maximum noise level and the same patterns during the night, sleep is more fragmented by freight trains than by passenger or automotive trains [99].

The association between annoyance or sleep disturbance and noise was found among residents. Especially in subjects exposed at higher noise levels [100]. Airport noise interferes with the quality of sleep of the people living near airports [101,102,103], as evidenced by some studies which showed that airport noise is associated with an increase in the frequency of sleeping pills and tranquilizer usage [73,104,105,106]. In addition, noise can activate the sympathetic and endocrine systems [6], with relative consequences on the psychological sphere [107,108,109]. Research on the relationship between annoyance and psychological health started many years ago. Psychological distress is often measured with the General Health Questionnaire (GHQ) in the different articles with the results being controversial [110]. Some authors did not find any significant association between aircraft noise exposure and psychological ill-health based on the GHQ-30 [111], the GHQ-28 [112] or the GHQ-12 [113]. Only studies in Japan and in Spain have shown a significant correlation between aircraft noise exposure and moderate/severe somatic symptoms identified by the GHQ-28 and GHQ-12 in people sensitive to noise [112,114]. High noise sensitivity was identified by Stansfeld et al. [115], as a predictor of psychological distress using the GHQ-30. It is often also necessary to consider the reciprocal relationship between the different factors. Thus, extremely annoyed people can develop psychological ill health but it is also possible to have an opposite effect, with annoyance symptoms manifesting in affected people [116,117]. In this review annoyance was also found to be dependent on psychological factors. Thus, noise sensitivity, distance to the source, window opening behavior, bedroom orientation and position, degree of urbanization, sleep timing, sleep medication intake, survey season and night air temperature have all been implicated. Noise sensitivity is considered as a moderating factor of the effects of aircraft noise exposure on annoyance [118,119,120,121]. This variable could also influence the effects of noise on psychological ill-health [122]. Noise sensitivity is a potential indicator of vulnerability to environmental stressors [123,124], such as a proxy measure of anxiety and irritation [116]. These individual factors are also involved in the newer sources of noise. For example, as reported in international literature when wind turbines are placed in residential areas, they can cause annoyance [125,126,127]. The visual impact of wind turbines is more pronounced in rural areas when compared to more densely populated areas [128] and among respondents that benefited economically from wind turbines the proportion of people who were rather or very annoyed was significantly lower [129,130,131].

Finally, we have noticed how in a certain number of works the authors are looking for cardiovascular and psychological disorders at the same time. This is an interesting aspect, as it is probably possible to hypothesize a synergy between the two areas or at least a mutual cause–effect correlation, as emerges from more recent studies [132,133,134,135].

This review has some limitations. Firstly, most of the studies are cross-sectional, not trials or efficacy evaluations, which would be of particular interest to the researches so as to understand the determinants of occupational diseases and to set up appropriate interventions. Among the publications included in his review, there is a high level of heterogeneity both in terms of number of exposed subjects (some research concerns a limited number of residents) as well as length of exposure (from a few months to many years for others). It was also very complex to compare the various different studies, carried out in environmental contexts, with very different cultures, religions and legislations.

## 5. Conclusions

Considering the constantly growing trend of new sources of noise and the particular susceptibility of people, caused by numerous factors, it is becoming increasingly urgent to define the extent of noise exposure, its severity and the correlation between sound input and the deterioration of the quality of life caused in the population. In 2005, the European Commission dedicated the European Week on Workplace Health and Safety to noise, developing numerous information and communication initiatives aimed at raising public awareness of this risk agent. In order to address the problem of environmental noise with long-lasting solutions, it is therefore necessary to quantify the effects of external noise, either to predict new socio-economic impacts or in relation to the health of residents, to develop new policy strategies and finally, to create new guidelines. These should aim at easing the severity of the problem and avoiding complications in the medium to long term. In order to do this, it is clear that socio-acoustic surveys are an indispensable tool for standardizing the correlation between noise reactivity and the extent of provocative noise.

## Figures and Tables

**Figure 1 ijerph-17-06621-f001:**
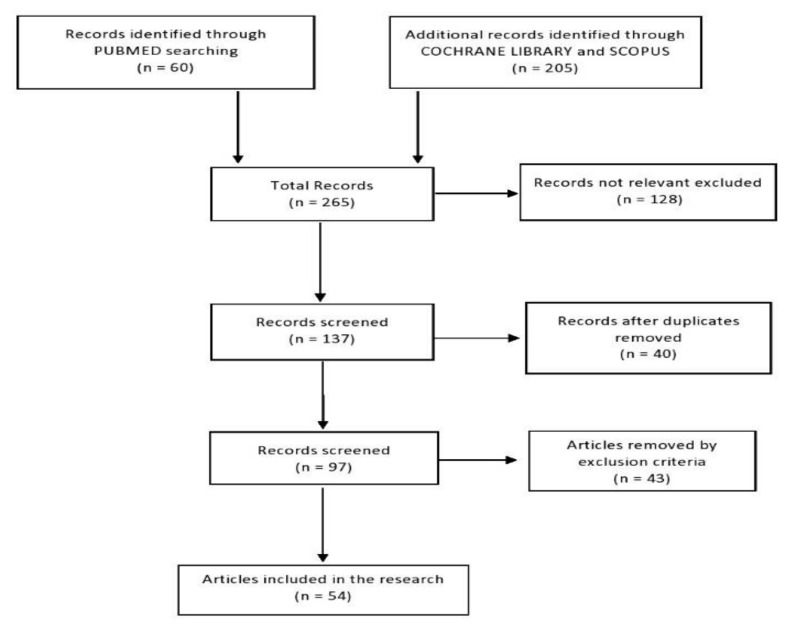
Flow chart of the systematic review.

**Table 1 ijerph-17-06621-t001:** Tools for assessing the quality of studies included in this systematic review.

Scale	Examined Study	Questions	Scores Range
Amstar	Systematic Reviews	N.11 (*yes, no, cannot answer, not applicable*)	0–11 pt
Insa	Narrative Reviews	N.7 (*yes, no*)	0–7 pt
Jadad	Randomized Trials	N.3 (*yes, no*)	0–5 pt
New Castle Ottawa	Case- Control	Selection N.4, Comparability N.1, Exposure N.3 (*yes/no*)	0–8 pt
New Castle Ottawa	Cross sectional	Selection N.4, Comparability N.1, Outcome N.2 (*yes/no*)	0–10 pt
New Castle Ottawa	Cohort Studies	Selection N.4, Comparability N.1, Outcome N.3 (*yes/no*)	0–8 pt

**Table 2 ijerph-17-06621-t002:** All the studies included in this systematic review, in alphabetical order.

First Author	Year	Study	Country	Noise Exposure	Disease
Ancona	2014	cross sectional	Italy	airport	sleep disturbance, annoyance, cardiovascular
Bakker	2012	cross sectional	Netherlands	wind turbine	annoyance, sleep disturbance
Baudin	2018	cross sectional	France	airport	annoyance, psychological health
Brink	2019	cross sectional	Switzerland	road, rail, airport	sleep disturbance
Brink	2019	cross sectional	Switzerland	road, rail, airport	Annoyance
Brown	2015	cross sectional	China	road traffic	sleep disturbance
Bunnakrid	2017	cross sectional	Thailand	road traffic	Annoyance
Camusso	2016	cross sectional	Italy	road traffic	Annoyance
Elmehdi	2012	cross sectional	Emirates	airport	Annoyance
Elmenhorst	2019	trial	Germany	road, rail, airport	sleep disturbance
Erikson	2017	cross sectional	Sweden	road, rail	sleep disturbance, annoyance, cardiovascular
Fryd	2016	cross sectional	Denmark	road traffic	Annoyance
Gjestland	2017	cross sectional	Norway	airport	Annoyance
Gjestland	2015	cross sectional	Vietnam	airport, road	Annoyance
Gjestland	2019	cross sectional	Norway	airport, road	Annoyance
Guski	2017	Systematic review	Germany	airport, road, railway	Annoyance
Hays	2016	narrative review	USA	oil gas development	sleep disturbance, annoyance, cardiovascular
Hong	2010	cross sectional	Korea	road, rail	sleep disturbance
Hongisto	2017	cross sectional	Finland	wind turbine	Annoyance
Hume	2010	narrative review	Uk	airport	sleep disturbance
Janssen	2011	cross sectional	Sweden, Netherlands	wind turbine	Annoyance
Kageyama	2016	case control	Japan	wind turbine	sleep disturbance
Kim	2014	case control	Korea	airport	sleep disturbance
Kim	2012	cross sectional	USA	road traffic	annoyance, sleep disturbance
Lercher	2013	cross sectional	Austria	road traffic	Annoyance
Lechner	2019	cross sectional	Austria	road, rail, airport	Annoyance
Lercher	2011	narrative review	Austria	road, rail	cardiovascular, annoyance
Lercher	2017	cross sectional	Austria	road, rail	Annoyance
Lercher	2012	cross sectional	Austria	road, rail, airport	annoyance, sleep disturbance
Lercher	2010	cross sectional	Austria	rail	sleep disturbance
Liu	2017	cross sectional	China	construction	Annoyance
Magari	2014	cross sectional	USA	wind turbine	sleep disturbance
Matsui	2013	cross sectional	Japan	airport	psychological distress
Miller	2015	cross sectional	USA	airport	Annoyance
Morinaga	2016	cross sectional	Japan	airport	Annoyance
Muller	2016	cohort study	Germany	airport	sleep disturbance
Ogren	2017	cross sectional	Sweden	rail	Annoyance
Pedersen	2015	cross sectional	Sweden	road traffic	Annoyance
Pennig	2014	cross sectional	Germany	rail	Annoyance
Poulsen	2019	cohort study	Denmark	wind turbine	sleep disturbance
Ragettli	2015	cross sectional	Canada	road, rail, airport	Annoyance
Schmidt	2015	trial	Germany	airport	cardiovascular, sleep disorders
Schmidt	2014	Systematic review	Denmark	wind turbine	annoyance, sleep disorders
Schreckenberg	2013	cross sectional	Germany	rail	Annoyance
Schreckenberg	2016	cohort study	Germany	airport	annoyance, sleep disturbance
Schreckenberg	2010	cross sectional	Germany	airport	Annoyance
Shepherd	2013	cross sectional	New Zealand	wind turbine, airport	Annoyance
Shimoyama	2014	cross sectional	Japan	road traffic	annoyance, sleep disturbance
Silva	2016	cross sectional	Brazil	airport	Annoyance
Tainio	2015	cross sectional	Poland	road traffic	Annoyance
Tobollik	2019	cross sectional	Germany	road, rail, airport	sleep disturbance, annoyance, cardiovascular
Trieu	2019	cross sectional	Japan	airport	sleep disturbance, annoyance, cardiovascular
Wothge	2017	cross sectional	Germany	road, rail, airport	Annoyance
Yano	2013	cross sectional	Japan	wind turbine	Annoyance

**Table 3 ijerph-17-06621-t003:** Reviews included with their relative score.

First Author	Included Articles	Principal Results	Score
Guski	62	The evidence of exposure–response relations between noise levels and % HA is moderate (aircraft, railway) or low (road traffic, wind turbines). The evidence of correlations between noise levels and annoyance raw scores is high (aircraft, railway) or moderate (road traffic, wind turbines)	A.8
Hays	narrative	oil and gas activities produce noise at levels that may increase the risk of adverse health outcomes, including annoyance, sleep disturbance, and cardiovascular diseases	I.5
Hume	narrative	annoyance is the mediating factor between noise exposure and cardiovascular diseases with annoyance has associations with a number of cofactors such as noise sensitivity, negative affectivity and mental health	I.6
Lercher	narrative	important modifiers may partly be responsible for the large variations found in the noise health effects (socio-demographic factors, length of exposure, bedroom.)	I.6
Schmidt	36	a dose–response relationship between wind turbine noise linked to noise annoyance, sleep disturbance and possibly even psychological distress is present in the literature	A.6

**Table 4 ijerph-17-06621-t004:** Cross articles included in this review, in alphabetical order, with their relative scores.

First Author	Included Subjects	Exposure Range	Questionnaire	Results	Scores
Ancona	N.896322	Lden 55–70 dB	not used	above 55 dB there were 4607 cases of hypertension,3.4 cases of AMI, 9789 cases of annoyance, 5084 sleep disorders	N.6
Bakker	N.725	21–54 dB	GHQ	a dose–response relationship was found between wind turbine sound and annoyance	N.8
Baudin	N.1244	<45–>60 dB	GHQ	22% of the participants were considered to have psychological ill-health; annoyance due to aircraft noise and noise sensitivity were both significantly associated with psychological ill-health	N.8
Brink	N.5592	20–80 dB	ICBEN 5-point scale	bedroom orientation shows strong effect with sleep disorders	N.8
Brink	N.5592	Lden 30–85 dB	ICBEN 11-point scale	aircraft noise annoyance scored markedly higher than annoyance to railway and road traffic noise at the same Lden level. Railway noise elicited higher percentages of highly annoyed persons than road traffic noise.	N.8
Brown	N.10077	Lden 42–78 dB	Weinstein scale	population in Hong Kong exposed to high levels of road traffic noise (>70 dB) is similar to that found in cities in Europe. However, a much higher proportion of the population in Hong Kong compared to European cities is exposed to Lden levels of road traffic noise of 60–64 dB, and a much lower proportion to the lower levels (<55 dB).	N.7
Bunnakrid	N.253	Leq 69.3–75.4	ICBEN 5-point scale	average annoyance scores of traffic noise in Muang Phuket, Thalang, and Kathu were 1.78, 2.52, and 2.75; a significant positive correlation between road traffic noise and annoyance level (*p* = 0.025)	N.6
Camusso	N.830	Leq 35–105 dB	ICBEN 5,7 point- scale	people are more annoyed in broad streets than in narrow streets; dose–response curve shows a higher sensitivity in people living in broad street	N.7
Elmehdi	N.23	Ldn 40–80	ISO/TS 15666-2003	41% of the respondents near Dubai airport are highly annoyed	N.6
Erikson	N.971839	not specified	not used	DALY attributed to traffic noise in Sweden was estimated to be 36 711 (90%) related to road traffic and 4322 (10%) related to railway traffic, specially sleep disorders, 22 218 DALY (54%), followed by annoyance, 12 090 DALY (30%) and cardiovascular diseases, 6725 DALY (16%).	N.8
Fryd	N.6761	48–75 dB	ISO/TS 15666-2003	outdoor annoyance was higher for motorways than urban roads while the indoor annoyance was the same	N.7
Gjestland	N.32	<40–> 80 dB	not specified	at so-called LRC airports, the number of highly annoyed residents increases with an increasing amount of traffic. The same tendency cannot be found for HRC airports. At this type of airport the annoyance assessment is therefore most likely dominated by other non-acoustical factors	N.6
Gjestland	N.104	not specified	not used	the CTL method for characterizing the annoyance caused by long term exposure to noise is a robust method that segregates acoustical from non-acoustical influences on annoyance prevalence rates	N.7
Gjestland	N.7199	<40–> 80 dB	ICBEN 5-point scale	CTL was 73 dB for aircraft noise and 84 dB for road Noise	N.7
Hong	N.1160	LAeq 49–74 dB	CENVR	sleep is affected more by railway noise than by road traffic noise; sensitivity was shown to be a significant modifying factor	N.7
Hongisto	N.429	LAeq 26.7–44.2 dB	ISO/TS 15666-2003	indoor noise annoyance was correlated with sound level and distance (*p* = 2.4 × 10; *p* = 8.5 × 10)	N.7
Janssen	N.351, 754, 725	25–60 dB	ICBEN 5 point-scale	annoyance due to wind turbine noise is found at low exposure level; percentage of annoyance by wind turbine noise is expected at much lower levels of Lden than the same percentage of annoyance by for instance road traffic noise	N.7
Kim	N. 109967	<40–>80 dB	not specified	many residents of the greater Atlanta area may be exposed to noise levels that put them at risk of being highly annoyed or having high levels of sleep disturbance	N.6
Lercher	N. 2002/1643	<40–>80 dB	not specified	In Alpine valley, accumulation of factors can in some cases lead to higher annoyance from main roads than from highways	N.7
Lechner	N.1031	<45–>55 dB	ICBEN 11-point scale, EU-SILC 2015, LEF-K	all traffic noise sources positively and significantly increased the overall-annoyance score	N.8
Lercher	N.1641	<40–>80 dB	ICBEN 5 point-scale	distance to highway and railway track is negatively associated with annoyance (*p* < 0.001) while distance to the main road slightly failed significance (*p* = 0.071), sleep disturbance and coping scores are positively associated with higher annoyance (*p* < 0.001). Longer duration of living in the home is not significantly associated with higher annoyance (*p* = 0.163)	N.6
Lercher	not specified	<40–>80 dB	ICBEN 11-point scale	a linear dose–response relation was found between number of events >69 dBA and % rather and very annoyed.	N.6
Lercher	N.1643	40–75 dB	5-point Likert-type, PCL-C	more than twice the probabilities of medication intake at any level of railway sound exposure, in particular between 65–75 dB	N.7
Liu	N.1027	LAeq 15.30–77 dB	ICBEN 7,11 point- scale	when LAeq of construction noise increases from 60 dB to 80 dB, highly annoyed increase from 15% to 40%	N.6
Magari	N.62	not specified	Pedersen 2004	no statistically significant associations between sound level measurements inside or outside, and an individual’s assessment of their satisfaction with living environment and annoyance with the turbines at the P < 0.05 level	N.7
Matsui	N.3215	Lden 55–70 dB	Total Health Index	the PSD score showed significant association with sleep disturbance, although the annoyance score showed higher association with speech interference than sleep disturbance.	N.6
Miller	N.366	not specified	Not validated	those who believe the airport is very important are less likely to be annoyed by the noise.	N.5
Morinaga	N.4298	Lden 31–80 dB	ICBEN 5 point-scale	Lden value for military aircraft noise is 5–7 dB higher than civilian at an equal rate annoyance response	N.6
Ogren	N.1203	40.8–64.9 dB	ISO/TS 15666:2003	annoyance from noise may be influenced by the presence of vibration (*p* = 0.022)	N.6
Pedersen	N.385	not specified	GHQ	The highest frequencies of annoyance were found for vibration from buses or trucks (23%), noise from passing cars (22%), noise from mopeds and motorbikes (20%), motorway noise (17%)	N.6
Pennig	N.380	40–89.9 dB	ICBEN 11-point scale	64.3% are highly annoyed by trains and 20.7% by roads, especially during night	N.6
Ragettli	N.4336	50.1–76.1 for LAeq24h	European LARES- Survey	annoyed by road traffic, airplane and train noise was 20.1%, 13.0% and 6.1%, respectively	N.6
Schreckenberg	N.1211	<40–85 dB	ICBEN 5 point-scale	%HA and %HSD due to railway noise increases with increasing railway noise levels. For equivalent sound levels above 65 dB %HA for railway noise railway at daytime against L day is somewhat higher than %HA at night and considerably higher than %HSD against L night	N.6
Schreckenberg	N.2312	<40–>60 dB	Not validated	aircraft noise annoyance is associated with sound levels as well as with the number of flyovers (N55, N70). However, the strongest exposure–annoyance relationship for aircraft noise was found between the equivalent sound level and aircraft noise annoyance	N.6
Shepherd	N.823	Lden 55–76 dB	Whoqol-Bref, Noiseq	the dose–response relationships between noise annoyance and HRQOL measures indicated an inverse relationship; quiet areas were found to have higher mean HRQOL domain scores than noisy areas	N.6
Shimoyama	N.4966	Lden 61–83 dB, LAeq 50–73 dB	ICBEN 5,11 point- scale	dose–response curve showed that Vietnamese respondents were about 5 to 10 dB less annoyed by road traffic noise than those of EU and Japan	N.5
Silva	N.547	37.5–75 dB	ISO 15666:2003	in the range of 67.5–70 dB, 68.4% of the sample is highly annoyed (CTL 50% = 65.3 dB)	N.6
Tainio	not specified	not specified	not used	58000 DALYs in Poland, 44% due to air pollution and 46% due to noise	N.6
Tobollik	not specified	not specified	not used	highest burden was found for road traffic noise in Germany, with 75,896 DALYs	N.7
Trieu	N.755	Lden 38–76 dB	not validated	no significant association between hypertension and noise exposure but a a significant relationship between insomnia and nocturnal noise exposure	N.6
Wothge	N.4905	40–60 dB	ICBEN 5-point verbal scale	annoyance grows significantly with the increase of the LAeq,24 h of the aircraft noise and in combination of noise sources (airport + rail/roads)	N.7
Yano	N.747	26–50 dB	ICBEN 5-point verbal scale	when LAeq, n increased from 26 to 50 dB, annoyed gradually increased from 3 to 21, from 6 to 27 and from 25 to 48%, respectively. Annoyance rate depends on home location, temperature and wave sound	N.6

**Table 5 ijerph-17-06621-t005:** Experimental, case-control, cohort study, with their relative scores.

First Author	Included Subjects	Exposure Range	Questionnaire	Results	Length	Score
Elmenhorst	237	45–80 dB	Freiburger Persoenlichkeits Inventar	sound pressure levels increased in the order aircraft < road < railway noise, the awakening probability from road and railway noise being not significantly different (*p* = 0.988). At 70 dB SPL, it was more than 7% less probable to wake up due to aircraft noise than due to railway	4–13 nights	J.2
Kageyama	747 cases/332 controls	35–40 dB	THI	odds ratio of insomnia was significantly higher when the noise exposure level exceeded 40 dB, whereas the self-reported sensitivity to noise and visual annoyance with wind turbines were also independently associated with insomnia	2010–2012	N.6
Kim	871 cases/134 controls	<60–>80 WECPNL	PSQI, DASS	sleep disturbance was 45.5% in the control group, 71.8% in the low exposure group, 77.1% in high exposure (p 0.001)	2009–2011	N.6
Mueller	202	not specified	Polysomnography	by reducing nocturnal overflights, awakening decreased from 2.0 per night in 2011 to 0.8 per night in 2012	2011–2013	N.5
Poulsen	584891	<24–>42 dB	not specified	WTN of ≥42 dB was associated with a HR = 1.14 for sleep medications and 1.17 for antidepressants (compared to <24 dB)	1996–2003	N.6
Schmidt	60	36–49 dB	PSQI	nighttime aircraft noise markedly impairs endothelial function in patients with or at risk for cardiovascular disease.	any nights	J.3
Schreckenberg	9244–3508	36–61 dB	ICBEN 5-point scale	exposure response curve for aircraft annoyance after opening new runway depends on local changes in sound level	2011–2013	N.5

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
