# Peer review of "Urban Noise and Psychological Distress: A Systematic Review"

_ijerph, 2020, doi:10.3390/ijerph17186621_

Round 1

Reviewer 1 Report

Thank you for an opportunity to review this manuscript. Let me commend you on the amount of reviews you have conducted. I have several concerns that need to be addressed before the manuscript can be published. 1) You need to make it clear what the manuscript is intending to address, what other systematic reviews have not done? 2) How does it differ from what we already know? Perhaps, consider indicating from onset what scope of the reviews will focus on from onset. Have a section on literature review critically discussing what is known and use this point out the rationale for your study. I am also concerned that the reviews appear to have been inadequate or incomplete. For example, of the reviewed studies, mostly studies in developed Euro-Western contexts were included and limited if not none from developing countries were included. None of the reviewed studies were from African continent despite a large body of studies implemented in South Africa. Finally, I worried about the language. It requires extensive editing beginning with the abstract.

Author Response

Dear reviewer,

We were aware that annoyance is a widely recognized noise disorder but this review is intended to be an update of the last 10 years of studies, as it is part of a European "Life Monza Project" that, among the various purposes, investigates the phenomenon of the annoyance among the residents of a Monza'district.

We want to make it clear on this occasion, as reported in the methodology, that we have included studies from all countries and, if some countries do not appear, it is due to selection and exclusion criteria.

However, we have tried to improve the English language, images and bibliography.

We remain available for further clarification,

Best Regards

Reviewer 2 Report

General comments:

Interesting and important topic. However, the review has multiple grammatical issues that could benefit from editing. In addition more attention to the methodology will improve the quality of the review.

Introduction:

  • Line 43- please add a reference. 
  • The title does not match the contents of the paper:
    • The topic is noise and psychological distress- however there is very little information on the psychological distress in the introduction.
    • The aim of the project is to assess major pathologies associated with noise??

The introduction needs to be more informative on the expodsure and outcomes of noise exposure - focusing on psychological disorders and potential confounders.

Methods:

  • Was this project registered on Prospero?
  • what were the keywords for the outcomes (pathologies) - the authors should provide their detailed search terms as a supplementary table.
  • in page 3, line 95- what are "noise disease"?
  • How is annoyance defined and rated?
  • in line 90: what other diseases where assessed-its not clear?
  • Are studies from all countries included?
  • where both qualitative and quantitative studies included?
  • Where all types of study designs included- including cross sectional?
  • What was the data extraction process- please state the data fields that were extracted from each paper in the review.

  The methodology is lacking some detail. 

Results:

  • line 109/110- why were the other 43 excluded to get a total of 54 remaining. Please describe the reasons for the exclusion. Especially since the exclusion criteria is not very detailed e.g. ?number excluded because they were editorials etc. Can be added to the text or the prisma diagram.
  • Table 1- if the authors are looking at psychological problems- annoyance, sleep disorder, exp;lain why cardio-vascular disorders are included. The disease outcomes that are being assessed must be explained in the methodology.
  • Please describe the AMSTAR and other scoring systems- what do the scores mean (a brief table in the methods would add value)
  • What about the risk of bias- how was this assessed? what criteria was used? where are the results of the risk of bias?
  • A lot of grammatical errors throughout.
  • Tables and figures are very small- increase font size.

Discussion and conclusion

The discussion should be more a critical analyses. not clear how this review has improved the knowledge in this area. 

What are the recommendations based on this review.

  •  

Author Response

Dear reviewer,

we have corrected the introduction and methods. We have improved the English language, changed the tables to MS format and changed the entire bibliography. In addition, we also included a new table on the questionnaires used to evaluate the studies. Your corrections in the required text are highlighted in yellow.

Best regards

Reviewer 3 Report

     Today, there is a solid recognition that noise is hazardous to mental and physical health.  The World Health Organization has stated that noise is a health hazard.  Major studies, based on huge number of subjects, have linked noise to cardiovascular disorders.  Papers have been written noting the need to reduce the decibel levels long associated with hearing loss.  There are strong studies linking noise to adverse impact on children's language development, cognition and learning. Diminished quality of life has been noted in communities exposed to overhead jet and nearby traffic noise.  Yet, the majority of studies referenced in this article point to annoyance as the major impact. Noise impacts have gone beyond annoyance! 

     While it is true that additional studies can bring a greater understanding of the impacts of noise on health, the literature is plentiful enough today to call for greater actions to reduce noise pollution.  Yet, if the emphasis, as in this paper, is that noise is "annoying," such actions will be delayed. 

Author Response

Dear reviewer,

We had to focus our interest on the effects of noise on annoyance because this review is closely linked to the "Life Monza Project", in which precisely these aspects were investigated in a district of Monza city. For this reason, we decided to select and evaluate updates on the theme and the studies of the last 10 years. However, we have tried to improve the English language, images and bibliography.

We remain available for further clarification,

Best regards

Round 2

Reviewer 1 Report

Thank you for the amendments.

Although minor, there are still language errors. Vehicle traffic was fine, amend 'vehicular traffic'. 

Author Response

Thanks for the tips and advices. We have made the corrections.

Reviewer 2 Report

I am responding to the responses from the authors to my initial comments (yellow HIGHLIGHTs). The authors have tried to answer all my queries, however i feel they may have misunderstood some comments: 

1. How is annoyance defined and rated?  

Author response:it is not clear where we need to enter this definition

Reviewer- should go in the introduction or methodology. Please add.

2.Where all types of study designs included- including cross sectional?

Author-, line 104

Reviewer: move this statement to the inclusion criteria. Since its not an exclusion criteria.

3.What was the data extraction process- please state the data fields that were extracted from each paper in the review

Author: Done, line 95-96

Reviewer: the authors have not understood- I mean what data points or variables were looked for in each paper- e.g. date of publication, country, population size, sample size, gender distribution etc- Did they have a data extraction sheet to enter data for each paper. If yes- what were the headings/ variables in this data collection sheet. This must be added to the paper.

4. line 109/110- why were the other 43 excluded to get a total of 54 remaining. Please describe the reasons for the exclusion. Especially since the exclusion criteria is not very detailed e.g. ?number excluded because they were editorials etc. Can be added to the text or the prisma diagram.

Author: Done, Line 123 -124.

Reviewer: It is not enough to say “based on exclusion criteria- especially since the exclusion criteria are not specific. You need to state x number of articles excluded because there was no quantitative data etc. A systematic review needs to be as precise as possible.

5. Table 1- if the authors are looking at psychological problems- annoyance, sleep disorder, explain why cardio-vascular disorders are included. The disease outcomes that are being assessed must be explained in the methodology.

author: Cardiovascular disease includes only if mentioned in the selected articles, together with annoyance and sleep disorders.

REviewer: What is the purpose or need to highlight this in the discussion, unless it specifically relates to the topic? Please adjust accordingly or explain how cardiovascular disease relates to your exposure and outcome (psychological distress).

6. What about the risk of bias- how was this assessed? what criteria was used? where are the results of the risk of bias? 

author:Done, lines 110, 121-

reviewer: This is not risk of bias- I am talking about the risk of bias in the articles included in the study. This has to be done. Please assess each paper for possible risk of bias- there are tools available – look for selection bias, misclassification of bias of exposure, misclassification bias of the outcome, publication bias, bias due to missing data etc. A heat map can be created as well. This assessment is then included in the quality assessment which asses other characteristics as well.

Author Response

thanks for the tips and advice.
We have made the required corrections, for example added the definition of annoyance, calculated risk of bias with RobVis, better clarified the data extraction-article selection process.
corrections are underlined in green in the text.  

Reviewer 3 Report

The authors responded to my earlier comments in that their paper focused only on annoyance because this was the response explored by the Life Monza  undertaking.  (Noticed that the funding for this paper came from this project).  However, there was no discussion within the text of the paper that stated that the project was concerned only with the "annoyance" response to noise. It is my understanding that the Life Monza project was interested in reducing noise pollution and I would assume the studies linking noise to adverse mental and physical impacts would add weight to the goals of the project.        Yes, English language, images and bibliography were improved.  Do reread: standardize is American version of spelling, not standardise..         Thus, my earlier evaluation of this paper should hold. 

Author Response

Thanks for the tips and advices. We have made some corrections.
